# Secnidazole Is a Promising Imidazole Mitigator of *Serratia marcescens* Virulence

**DOI:** 10.3390/microorganisms9112333

**Published:** 2021-11-11

**Authors:** Ahdab N. Khayyat, Hisham A. Abbas, Maan T. Khayat, Moataz A. Shaldam, Momen Askoura, Hani Z. Asfour, El-Sayed Khafagy, Amr S. Abu Lila, Ahmed N. Allam, Wael A. H. Hegazy

**Affiliations:** 1Department of Pharmaceutical Chemistry, Faculty of Pharmacy, King Abdulaziz University, Jeddah 21589, Saudi Arabia; ankhayyat@kau.edu.sa (A.N.K.); mkhayat@kau.edu.sa (M.T.K.); 2Department of Microbiology and Immunology, Faculty of Pharmacy, Zagazig University, Zagazig 44519, Egypt; hishamabbas2008@gmail.com (H.A.A.); momenaskora@yahoo.com (M.A.); 3Department of Pharmaceutical Chemistry, Faculty of Pharmacy, Kafrelsheikh University, Kafr El-Sheikh 33511, Egypt; dr_moutaz_986@yahoo.com; 4Department of Medical Microbiology and Parasitology, Faculty of Medicine, King Abdulaziz University, Jeddah 21589, Saudi Arabia; hasfour@kau.edu.sa; 5Department of Pharmaceutics, College of Pharmacy, Prince Sattam Bin Abdulaziz University, Al-kharj 11942, Saudi Arabia; e.khafagy@psau.edu.sa; 6Department of Pharmaceutics and Industrial Pharmacy, Faculty of Pharmacy, Suez Canal University, Ismailia 41552, Egypt; 7Department of Pharmaceutics, College of Pharmacy, University of Hail, Hail 81442, Saudi Arabia; a.abulila@uoh.edu.sa; 8Department of Pharmaceutics and Industrial Pharmacy, Faculty of Pharmacy, Zagazig University, Zagazig 44519, Egypt; 9Department of Pharmaceutics, Faculty of Pharmacy, Alexandria University, Alexandria 21521, Egypt; allam@alexu.edu.eg

**Keywords:** *Serratia marcescens*, quorum sensing, secnidazole, anti-virulence agents, anti-quorum sensing agents, virulence factors, pathogenesis

## Abstract

*Serratia marcescens* is an opportunistic pathogen that causes diverse nosocomial infections. *S. marcescens* has developed considerable resistance to different antibiotics and is equipped with an armory of virulence factors. These virulence factors are regulated in *S. marcescens* by an intercellular communication system termed quorum sensing (QS). Targeting bacterial virulence and QS is an interesting approach to mitigating bacterial pathogenesis and overcoming the development of resistance to antimicrobials. In this study, we aimed to evaluate the anti-virulence activities of secnidazole on a clinical isolate of *S. marcescens*. The effects of secnidazole at sub-inhibitory concentrations (sub-MICs) on virulence factors, swarming motility, biofilm formation, proteases, hemolysin activity, and prodigiosin production were evaluated in vitro. Secnidazole’s protective activity against *S. marcescens* pathogenesis was assessed in vivo in mice. Furthermore, a molecular docking study was conducted to evaluate the binding ability of secnidazole to the *S. marcescens* SmaR QS receptor. Our findings showed that secnidazole at sub-MICs significantly reduced *S. marcescens* virulence factor production in vitro and diminished its pathogenesis in mice. The *insilico* docking study revealed a great ability of secnidazole to competitively hinder the binding of the autoinducer to the SmaR QS receptor. In conclusion, secnidazole is a promising anti-virulence agent that may be used to control infections caused by *S. marcescens*.

## 1. Introduction

*Serratia marcescens* (family Enterobacteriaceae) is a motile facultative anaerobic Gram-negative rod that spreads abundantly in soil, plants, water, animals, and on surfaces [1]. *S. marcescens* was considered a non-pathogenic microbe until 1951 when it caused a nosocomial infection outbreak [2]. *S. marcescens* is a frequent opportunistic human pathogen that can cause several hospital-acquired infections, for instance, pneumonia, intravenous catheter-associated infections, endocarditis, osteomyelitis, and urinary tract infections [2,3]. It was reported in the USA as the tenth most frequent etiological agent of healthcare-associated blood infections and the seventh most common causative agent of nosocomial pneumonia [4]. Furthermore, *S. marcescens* developed significant resistance to various antibiotic groups, such as aminoglycosides, β-lactam, and fluoroquinolones; this characteristic greatly enhanced its pathogenesis [5,6].

*S. marcescens* pathogenesis is credited to a diverse panel of virulence factors, including biofilm formation; bacterial motility; and the production of several exo-enzymes, such as nuclease, protease, hemolysins, and lipase [1,7]. These virulence factors are operated under the control of a signaling system known as quorum sensing (QS). QS is considered a chemically encoded language that is utilized to orchestrate the expression of virulence genes, as it regulates the expression of several virulence factor-encoding genes [1,8,9]. QS controls numerous physiological functions in *S. marcescens*, such as swarming motility; biofilm formation; hemolytic activity; butanediol fermentation; and the production of extracellular enzymes, prodigiosin, biosurfactant, and antibiotics [1,10,11,12]. Controlling QS and virulence factors is an interesting option to mitigate bacterial pathogenesis, and as a consequence, it provides a great chance to conquer and eradicate bacterial infections [1,7,13]. In this context, several research groups have designed several approaches to target bacterial virulence and QS [7,13,14,15,16,17,18,19,20]. Drug repurposing is one of these approaches and aims to discover the uses of already approved drugs other than their known clinical uses. This strategy confers several advantages and decreases costs and time [21].

Imidazole nucleus has been employed extensively to develop a diverse array of antimicrobial agents, including antibacterial, antifungal, and antiprotozoal agents, as reviewed in [22]. Recently, we investigated the anti-virulence and anti-QS activities of metronidazole, which harbors the imidazole moiety, against *Proteus mirabilis* isolated from macerated diabetic foot ulcers. Metronidazole showed superior efficiency in diminishing the virulence factors of multi-resistant *Proteus mirabilis*. [18]. These findings encouraged us to investigate the anti-virulence activities of another member of imidazoles, namely secnidazole. Secnidazole, which shares the 5-nitroimidazole nucleus with metronidazole, is clinically used to treat protozoal infections, such as amoebiasis and giardiasis, and bacterial vaginitis [23]. Moreover, secnidazole was shown to act as an analog of acylhomoserine lactones and effectively inhibited QS resulting in the attenuation of *Pseudomonas aeruginosa* pathogenesis [24]. Bearing in mind these observations, this study aimed to explore the anti-virulence activities of secnidazole against *S. marcescens* pathogenesis in vitro and in vivo. 

## 2. Materials and Methods

### 2.1. Chemicals and Media 

Secnidazole was purchased from Sigma-Aldrich (St. Louis, MO, USA). All the used solvents and chemicals were of pharmaceutical grade. All used media—Mueller–Hinton (MH) broth and agar, Luria–Bertani (LB) broth and agar, and Tryptone soy broth (TSB)—were purchased from Oxoid (Hampshire, UK). 

### 2.2. Bacterial Strain

The used *S. marcescens* strain in this study was isolated from the endotracheal aspiration of an admitted patient in the intensive care unit at Zagazig University Hospital, Zagazig, Egypt. Alongside the biochemical identification of the isolated *S. marcescens*, its ribosomal proteins were identified using a Matrix-Assisted Laser Desorption/Ionization–Time of Flight (MALDI–TOF) mass spectrometry instrument, and the percentage of identity was 100% [7,14,19]. The bacterial strain was not isolated specifically for this study; the patient’s consent was obtained according to the applied routine protocols for pathological and microbiological examination at Zagazig University Hospital, Zagazig, Egypt, which are in complete compliance with the Helsinki declarations and without any risk, danger, or burden for the patient.

### 2.3. Minimum Inhibitory Concentration (MIC) Determination 

To determine the MIC of secnidazole, the broth microdilution method was employed according to the Clinical Laboratory and Standards Institute Guidelines (CLSI, 2015) [13,18]. The secnidazole solution was two-fold serially diluted using Mueller–Hinton broth to obtain dilutions ranging from 80 to 0.3125 mg/mL. The wells of a 96-well microtiter plate were filled with 100 µL aliquots of the different secnidazole dilutions. The *Serratia marcescens* strain was inoculated in Mueller–Hinton broth and incubated overnight at 37 °C. The resulting suspension was diluted with sterile saline to a turbidity equivalent to 0.5 McFarland standard and then diluted 1:100 with Mueller–Hinton broth to yield an approximate cell density of 1 × 10^6^ CFU/mL. Aliquots of 100 µL of the prepared suspension were added to all secnidazole dilutions, and the plate was incubated for 20 h at 37 °C. The minimum inhibitory concentration was calculated by observing the lowest concentration of the drug that inhibited the visible turbidity.

### 2.4. Effect of Secnidazole at Sub-MIC on the Growth of S. marcescens 

To avoid any expected effect of secnidazole on bacterial growth, *S. marcescens* was cultured in the presence or absence of secnidazole at sub-MIC (2 mg/mL) according to Nalca et al. [25]. The turbidities of fresh *S. marcescens* overnight cultures were adjusted to 0.5 McFarland standard. A standard inoculum size (1 × 10^8^ CFU/mL) was used to inoculate LB broth containing 2 mg/mL of secnidazole and control LB broth free from secnidazole. The optical densities were then measured at 600 nm after overnight incubation at 37 °C using a Biotek Spectrofluorimeter (Biotek, Shoreline, WA, USA). 

Secnidazole was used at a sub-inhibitory concentration (sub-MIC equal to 2 mg/mL) in all subsequent experiments to investigate its effect on *S. marcescens* virulence without an influence on its growth. It is worth emphasizing that the bacterial cultures were adjusted to an optical density OD600 of 0.4 (1 × 10^8^ CFU/ mL) in all subsequent experiments to normalize the obtained results. 

### 2.5. Biofilm Inhibition Assay

The biofilm formation capability of the tested *S. marcescens* strain was evaluated according to Stepanovic et al. [26]. Briefly, 200 μL aliquots from *S. marcescens* (1 × 10^6^ CFU/mL at OD600 0.4) fresh overnight cultures were inoculated overnight in sterile 96-microtiter plates at 37 °C. The non-adherent cells were aspirated, and the adherent cells were fixed with methanol (95%) for 20 min at room temperature. The adhered cells were simply stained with crystal violet (1%) for 30 min. After washing out the excess stain, the bound dye was dissolved with 95% ethanol, and the optical density was measured at 590 nm. The ratio of the obtained OD of the dissolved dye bound to adhered cells and the cutoff OD (ODc) were calculated. The bacterial strain biofilm-forming capability was categorized as strong (OD > 4 × ODc), moderate (4 × ODc ≥ OD > 2 × ODc), weak (2 × ODc ≥ OD > ODc), or not biofilm-forming (OD ≤ ODc). 

For the evaluation of the inhibitory effect of secnidazole on biofilm formation, the same steps that were used to assess biofilm formation were repeated but in the presence of secnidazole. *S. marcescens* aliquots (100 μL) were added to the wells of 96-microtitre plates containing 100 μL of secnidazole at sub-MIC. The absorbances of the dissolved bound crystal violet to the adhered cells in the presence or absence of secnidazole at sub-MIC were measured at 590 nm. Secnidazole’s inhibitory effect on biofilm formation was calculated as a percentage change from the control [7,20,27,28]. 

### 2.6. Assay of S. marcescens Motility Inhibition

The inhibitory ability of secnidazole against *S. marcescens* swarming motility was assessed as described earlier [13,17,18]. Briefly, a fresh overnight culture of *S. marcescens* was suspended in LB broth, and the optical density of bacterial growth was adjusted to OD 0.4. Five microliters from the prepared suspension was inoculated into the center of LB agar plates containing secnidazole (2 mg/mL) and control plates. After overnight incubation at 28 °C, the swarming zones were measured. 

### 2.7. Protease Inhibition Assay

The semi-quantitative skim milk agar method was employed to quantify the production of protease in the presence and absence of secnidazole at sub-MIC as described previously [7,29]. Briefly, fresh *S. marcescens* cultures were grown overnight in LB broth in the presence or absence secnidazole at sub-MIC at 37 °C. The supernatants were collected by centrifugation, and 100 μL aliquots were added to the wells of skim milk agar plates. After overnight incubation at 37 °C, the diameters of the clear zones surrounding the wells were measured in mm. The inhibition of protease production by secnidazole at sub-MIC was calculated as a percentage change from untreated bacteria as shown previously [7]. 

### 2.8. Prodigiosin Inhibition Assay

The prodigiosin production by *S. marcescens* in the presence or absence of secnidazole was quantified as described earlier [7,13,14,19]. The optical densities of suspensions prepared from fresh *S. marcescens* cultures were adjusted to 1 × 10^6^ CFU/mL at an OD of 0.4 and inoculated into LB broth with or without secnidazole at sub-MIC. After 20 h of incubation at 37 °C, the cells were collected by centrifugation, and the prodigiosin was extracted by 4% 1 M HCl in ethanol. The absorbance was measured at 534 nm, and the inhibitory effect of secnidazole on prodigiosin production was calculated as a percentage change from the untreated control.

### 2.9. Hemolysis Inhibition Assay

For the assay of hemolysin, *S. marcescens* isolate was cultured in LB broth containing or not containing secnidazole at sub-MIC at 37 °C overnight, and the supernatants were collected by centrifugation [17,19]. Briefly, 2% erythrocyte suspensions (obtained from experimental animals) in sterile saline (0.8 mL) were mixed with 0.5 mL of bacterial supernatants and incubated at 37 °C. After 2 h, the mixtures were centrifuged, and the absorbances were measured at 540 nm. The obtained measures were compared with the absorbances of the negative control (un-hemolyzed erythrocytes) and the positive control of hemolyzed erythrocytes, which was prepared by adding 0.1% sodium dodecyl sulfate. The inhibitory effect of secnidazole at sub-MIC on hemolysis activity was assessed as a percentage change from the untreated bacterial control. 

### 2.10. In Silico Docking of Secnidazole into the S. marcescens SmaR QS Receptor

In this work, secnidazole and the co-crystallized natural ligand C4HSL were docked into the active site of our prior model for the SmaR protein [13] using AutoDock Vina [30]. Ligand structures were drawn into Marvin Sketch v.18.23.0 (Marvin Sketch, v.19.12, ChemAxon), and the most energetically favored conformer was exported as a (*.pdb) file. The AutoDockTools package [31] was used to assign Gasteiger atomic partial charges, and all the rotatable bonds in the ligands were set to be flexible. For receptor preparation, hydrogen atoms were added, Gasteiger atomic partial charges were assigned, and all receptors and ligands were converted to the PDBQT format using the AutoDockTools package for the docking process. In the AutoDock Vina configuration files, the parameter num modes was set to 10, and exhaustiveness was set to 10. The grid box center (*x* = 20.67, *y* = 20.59, and *z* = 20.06) with a size (*x* = 13, *y* = 13, and *z* = 13) was used to define the active site for docking. AutoDock Vina was executed. Pymol (PyMOL Molecular Visualization System, v.2.0, Schrödinger, New York, NY, USA) was used for 3D visualization, and the 2D schematic presentation was generated using LigPlot+ v.1.4.5 (European Bioinformatics Institute, Cambridgeshire, UK) [32].

### 2.11. In-Vivo Mice Protection Assay

The mouse in vivo survival model was applied to evaluate the secnidazole protective activity against *S. marcescens* pathogenesis as previously described [7,13,33]. Briefly, the optical densities of a fresh *S. marcescens* overnight culture in LB broth containing or not containing secnidazole at sub-MIC, along with LB broth with dimethyl sulfoxide (DMSO) at the same concentration that was used as a solvent, were adjusted to OD 0.4 (≈1 × 10^8^ CFU/mL) in phosphate-buffered saline (PBS). Female healthy albino mice (Mus musculus, 3 weeks old) were divided into five groups (*n* = 10) as follows: one group intraperitoneally injected with 100 μL of secnidazole-treated *S. marcescens* in sterile PBS; two positive groups, one that was injected with 100 μL of untreated *S. marcescens* and another that was injected with 100 μL of DMSO-treated *S. marcescens*; and two negative groups, one that was left uninoculated and another that was injected with 100 μL of sterile PBS. The survival of the mice in each group was documented daily over the experimental period (5 days) and plotted using the Kaplan–Meier method.

### 2.12. Statistical Analysis

All the performed experiments were done in triplicate, and the results are presented as the means ± standard error. The Student’s *t*-test was employed to evaluate significance, where a *p* value < 0.05 was considered significant (GraphPad Prism Software, v.8, San Diego, CA, USA).

## 3. Results

### 3.1. Determination of Secnidazole MIC and Its Effect on S. marcescens Growth 

Secnidazole inhibited *S. marcescens* growth at 10 mg/mL. The secnidazole anti-virulence activity was evaluated at 2 mg/mL (equivalent to 1/5 MIC). To exclude any effect of secnidazole on *S. marcescens* growth, the optical densities of the bacterial suspensions were measured in the absence or presence of secnidazole (2 mg/mL) at 600 nm. There was no significant difference between the bacterial suspension turbidities in the absence or presence of secnidazole at sub-MIC (*p* = 0.5946), which emphasizes that secnidazole has no inhibitory effect on *S. marcescens* growth.

### 3.2. Secnidazole Inhibits Biofilm Formation

To evaluate biofilm production, the absorbance of crystal violet of biofilm-forming *S. marcescens* was measured, and it was found to be 4 times greater than the cutoff value, indicating that the *S. marcescens* isolate is strongly biofilm-forming [7,14]. Furthermore, biofilm formation was quantified in the absence and presence of secnidazole at sub-MIC. Secnidazole produced a statistically significant reduction in biofilm formation of about 60% (Figure 1).

### 3.3. Secnidazole Interferes with Swarming Motility

The diameters of *S. marcescens* swarming on LB agar plates without or with 2 mg/mL of secnidazole were measured. Secnidazole significantly reduced the swarming motility by about 65% (Figure 2).

### 3.4. Secnidazole Diminishes S. marcescens Virulence Factors

The secnidazole inhibitory effect on *S. marcescens’* production of protease was assessed using the skim milk agar method. Secnidazole at sub-MIC significantly decreased the protease production by about 22% (Figure 3). Furthermore, the hemolysin activity of *S. marcescens* in the absence or presence of secnidazole at sub-MIC was evaluated in comparison with the negative control (non-hemolyzed erythrocytes) and the positive control (completely hemolyzed erythrocytes). Secnidazole at sub-MIC could significantly decrease the hemolytic activity of *S. marcescens* to about 44% compared with the untreated control (Figure 3). 

### 3.5. Secnidazole Reduces Prodigiosin Production

The prodigiosin produced by *S. marcescens* was quantified in the absence and presence of secnidazole at sub-MIC. Secnidazole showed a significant ability to reduce the production of the QS-controlled prodigiosin pigment by *S. marcescens* to about 71% (Figure 4).

### 3.6. Secnidazole Hinders the Binding of Autoinducer to S. marcescens QS Receptor

The binding mode of secnidazole to the SmaR protein inhibitor was revealed by the molecular docking study. The binding interactions of secnidazole and C4-HSL with the target receptor are shown in Figure 5. Secnidazole could bind by a wide range of interactions, including hydrogen bonding, hydrophobic, and electrostatic interaction. The autodock scores for each ligand, in addition to the interacting residues, are shown in Table 1. Furthermore, secnidazole lacks the hydrophobic tail that is present in the natural ligand, and this might impart the inhibitory mode of binding to secnidazole.

### 3.7. Secnidazole Protects Mice against S. marcescens In Vivo

The secnidazole protective activity against *S. marcescens* virulence was assessed in vivo. Five mouse groups, each comprised of 10 mice, were intraperitoneally injected with treated or untreated *S. marcescens*, and the dead animals in each group were recorded during the 5 days of the experiment. All mice survived in the negative control groups (PBS or uninfected), while only 6 out of 10 mice survived in the positive control groups (untreated or DMSO-treated *S. marcescens*). On the other hand, 9 out of 10 mice survived of those that were injected with secnidazole-treated *S. marcescens*, indicating that secnidazole conferred about 30% protection in comparison with mice injected with untreated bacteria (Figure 6). The treatment of *S. marcescens* with secnidazole at sub-MIC significantly diminished the bacteria’s capacity to kill mice (*p* = 0.0023) according to the log-rank test for the trend.

## 4. Discussion

*S. marcescens* is an opportunistic pathogen that is armed with an arsenal of diverse virulence factors. In addition to these virulence factors, the increasing capability of *S. marcescens* to develop resistance to several antibiotics constitutes an additional burden [6,34]. QS modulates the physiological functions in *S. marcescens* and regulates the expression of its virulence factors, as reviewed in [1]. Bacterial resistance development is one of the top risks, and hence overcoming such resistance is a priority [35]. This can be accomplished by identifying new strategies to conquer bacterial infections. Targeting virulence factors controlled by QS and/or QS itself may offer an interesting approach that possesses several merits [33,36,37]. This approach does not lead to complete eradication of bacteria, which leads to (i) the stimulation of the immune system for their complete eradication and (ii) a decrease in the pressure on bacterial growth, leading to the avoidance of resistance development [7,33]. In this direction, several chemical and natural compounds have been screened for their anti-virulence and anti-QS activities [13,15,17,18,19,20,33,37,38,39,40]. Furthermore, the anti-virulence activities of several approved drugs have been screened in order to repurpose them as anti-virulence and anti-QS agents [7,14,16,18,27,39,41].

Imidazole moiety has been employed extensively to develop antifungal, antiprotozoal, and antibacterial drugs. The imidazoles’ capability to interact with microbial DNA, which results in the inhibition of microbial protein synthesis, is believed to be the main mechanism of their action [22,42]. In previous work, we showed that one member of nitroimidazoles, metronidazole, diminished the pathogenesis of *Proteus mirabilis* isolated from diabetic foot wounds [18]. Intriguingly, we aimed in the current study to explore the anti-virulence activities of another member of imidazoles, secnidazole, by challenging the pathogenesis of a nosocomial *S. marcescens* isolate.

Before the evaluation of the anti-virulence activities of secnidazole, and to exclude any effect on bacterial growth, *S. marcescens* was grown in the absence and presence of secnidazole at sub-MIC (1/5 MIC). It was found that secnidazole at sub-MIC had no influence on bacterial growth, and all the anti-virulence activities of secnidazole were assessed at the same sub-MIC. Bacterial motility is an important factor in their adhesion and biofilm formation. Bacterial motility greatly enhances the invasion of epithelial cells and facilitates bacterial spread and biofilm formation [43,44]. Furthermore, biofilm formation is decreased in non-motile bacteria [45]. This means that bacterial motility and bacterial biofilm formation ability both facilitate infection spread and enhance the development of antibiotic resistance [46]. Secnidazole at sub-MIC significantly reduced biofilm formation and swarming motility by 60% and 65%, respectively. Extracellular enzyme production plays a crucial role in *S. marcescens* virulence; *S. marcescens* produces several extracellular enzymes [1]. Protease cleaves several important proteins, such as albumin, casein, secretory components, and gelatin, and breaks immunoglobulins A and G, resulting in the facilitation of the spread of infection and suppression of the host defense [29,47]. Hemolysins’ cytotoxicity causes damage to host cells and inflammation and hinders neutrophils’ defense effect. *S. marcescens’* hemolytic activity is attributed to the pore-forming toxin ShlA, which is secreted by a two-component secretion system on cell-to-cell junctions [48,49]. Secnidazole at sub-MIC significantly reduced the production of protease and hemolysin activity of *S. marcescens* by 22% and 44%, respectively.

Although pigments that are produced by pathogenic bacteria are usually assumed to be essential virulence factors, the red pigment prodigiosin’s role in the *S. marcescens* pathogenesis is a matter of debate [50]. The QS-controlled *S. marcescens* red pigments prodigiosin and prodiginine are widely used as antimalarial, antiprotozoal, anticancer, insecticidal, bactericidal algicidal, immunosuppressives, and coloring agents [51]. However, the lack of evidence of the virulent activity of *S. marcescens’* red pigment, prodigiosin; its QS-controlled release, and the inhibition of its production indicate the diminishing of QS functions. Our findings showed that secnidazole at sub-MIC significantly reduced *S. marcescens’* production of prodigiosin by 71%.

The most predominant QS systems in Gram-negative bacteria are the LuxI/LuxR systems; *S. marcescens* employs one of the LuxR family members, SmaR. Several virulence factors of *S. marcescens,* including motility; biofilm formation; and the production of prodigiosin, antibiotics, and extracellular enzymes are controlled by the SmaI/SmaR QS system, as reviewed in [1]. *S. marcescens’* SmaR senses the N-acylhomoserine lactone analogs C4-HSL and C6-HSL [13,19]. For a more adequate understanding of how secnidazole diminishes the *S. marcescens* virulence factors, a molecular in silico docking study was performed to evaluate secnidazole’s ability to hinder the binding of the natural ligand to the SmaR QS receptor. Clearly, secnidazole could bind the SmaR receptor through several interaction modes, including hydrogen bonds, hydrophobic, and electrostatic interactions. These findings are in compliance with the previous results, which showed the inhibitory effects of secnidazole at sub-MIC on bacterial virulence. This leads us to propose that the anti-virulence activities of secnidazole are principally attributable to its ability to hinder QS receptors and compete with QS natural inducers.

To test the in vitro anti-virulence findings, a mouse survival in vivo model was used to assess secnidazole’s protection against *S. marcescens*. In compliance with the in vitro phenotypic findings, secnidazole at sub-MIC offered about 30% protection to mice, showing a trend for more survival compared with mice injected with untreated bacteria. In agreement with our data, secnidazole was shown to mitigate the *Pseudomonas aeruginosa* virulence and protect mice from its pathogenesis [24]. On the basis of these findings, treating *S. marcescens* with secnidazole at sub-MIC noticeably decreased the bacterial capacity to produce virulence factors both in vitro and, hence, bacterial host pathogenesis in vivo.

## 5. Conclusions

Targeting bacterial virulence and QS is a promising strategy to conquer bacterial pathogenesis, particularly when already approved safe drugs are repurposed. This strategy possesses several advantages; mainly, it is less likely to result in the emergence of bacterial resistance, and it enhances host immunity. This work offers a new use of secnidazole comprising the efficient diminishing of *S. marcescens* virulence factors, which might be conferred by suppressing the QS bacterial system. Secnidazole reduced *S. marcescens* biofilm formation; motility; and production of prodigiosin, protease, and hemolysins. Furthermore, secnidazole protected mice from *S. marcescens* in vivo. Our results suggest that secnidazole has potent anti-virulence and anti-QS activities, and herein, we present the application of secnidazole and other related imidazoles as an adjuvant therapy to traditional antibacterial agents to treat resistant bacterial infections after further pharmacological studies.

## Figures and Tables

**Figure 1 microorganisms-09-02333-f001:**
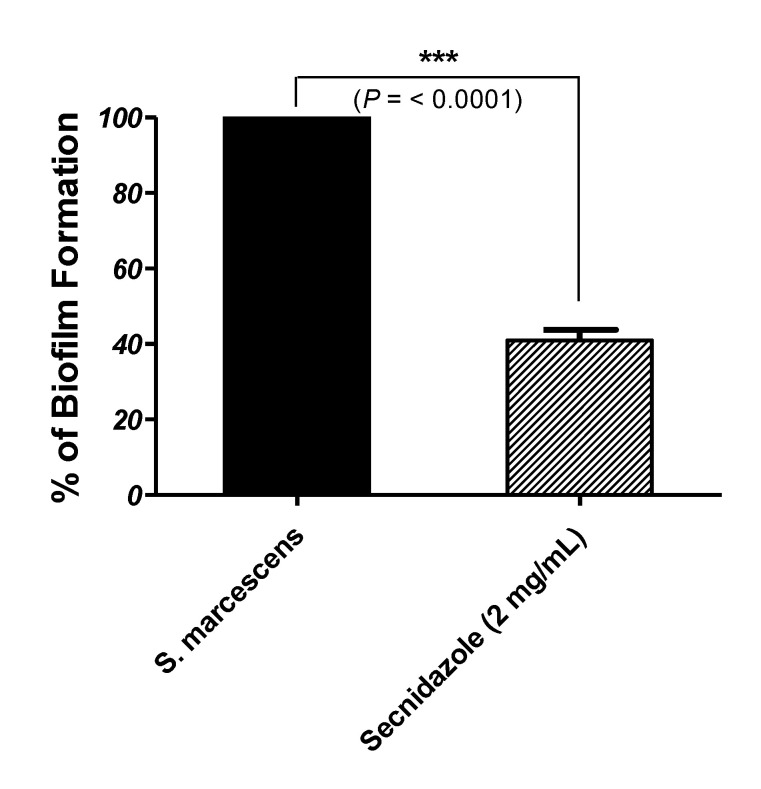
Biofilm inhibition of *S. marcescens* by secnidazole. The experiment was performed in triplicate, and the results are shown as the means ± standard error. Significance of mean difference between secnidazole-untreated and -treated *S. marcescens* was tested using the Student’s *t*-test on obtained optical densities values and the result was assumed statistically significant when *p* < 0.05. The results were shown as means ± standard error of biofilm production percent change by secnidazole in sub-MIC-treated *S. marcescens* from untreated *S. marcescens*. Secnidazole at sub-MIC significantly reduced biofilm production (*p* < 0.0001, *** = *p* < 0.001.

**Figure 2 microorganisms-09-02333-f002:**
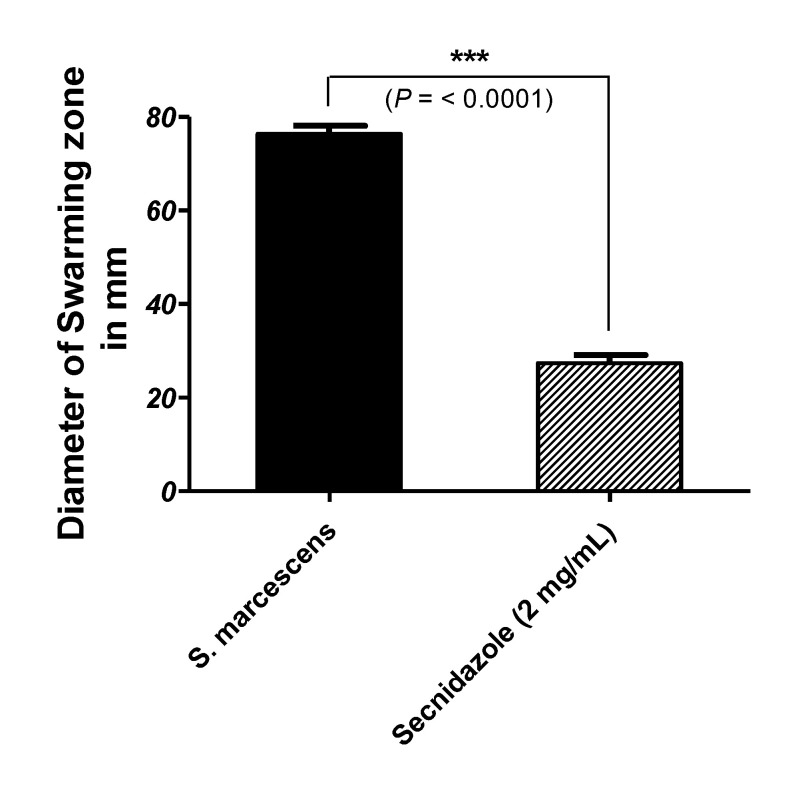
Inhibition of *S. marcescens* swarming motility by secnidazole. LB agar plates without or with 2 mg/mL secnidazole were prepared. Student’s *t*-test was used to compare *S. marcescens* not treated and treated with secnidazole at sub-MIC, and statistical significance was considered when *p* values were <0.05. The experiments were repeated in triplicate, and the results are presented as the means ± standard error. Secnidazole at sub-MIC significantly reduced *S. marcescens* swarming (*p* < 0.0001), *** = *p* < 0.001.

**Figure 3 microorganisms-09-02333-f003:**
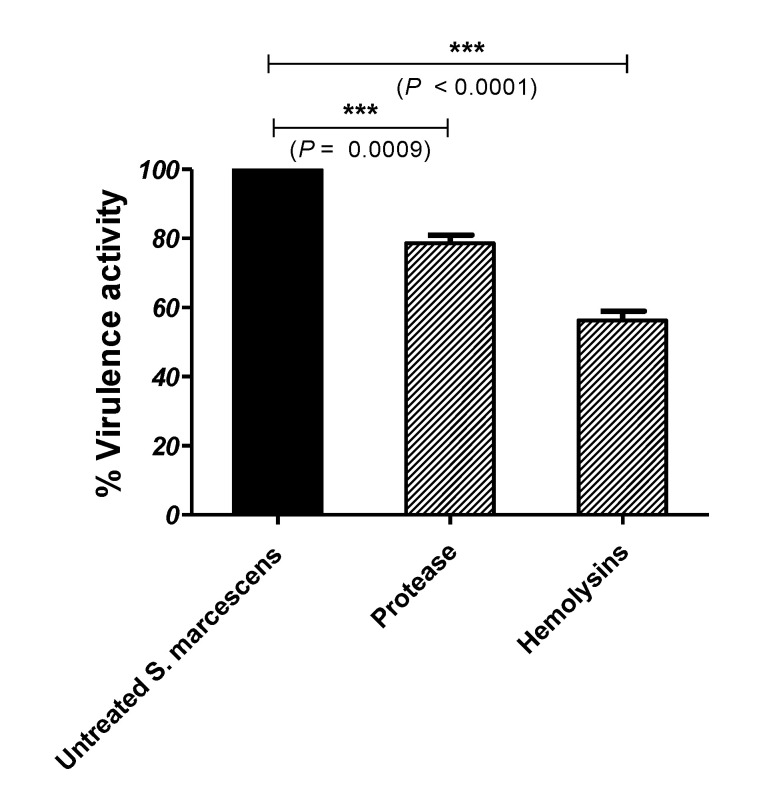
Inhibition of *S. marcescens’* protease and hemolysins activities by secnidazole. Student’s *t*-test was used to compare between *S. marcescens* not treated and treated with secnidazole at sub-MIC, and statistical significance was considered when *p* values were <0.05. The experiment was conducted in triplicate, and the results are shown as the means ± standard error. Secnidazole at sub-MIC significantly reduced the production of protease and hemolysins by *S. marcescens*, *** = *p* < 0.001.

**Figure 4 microorganisms-09-02333-f004:**
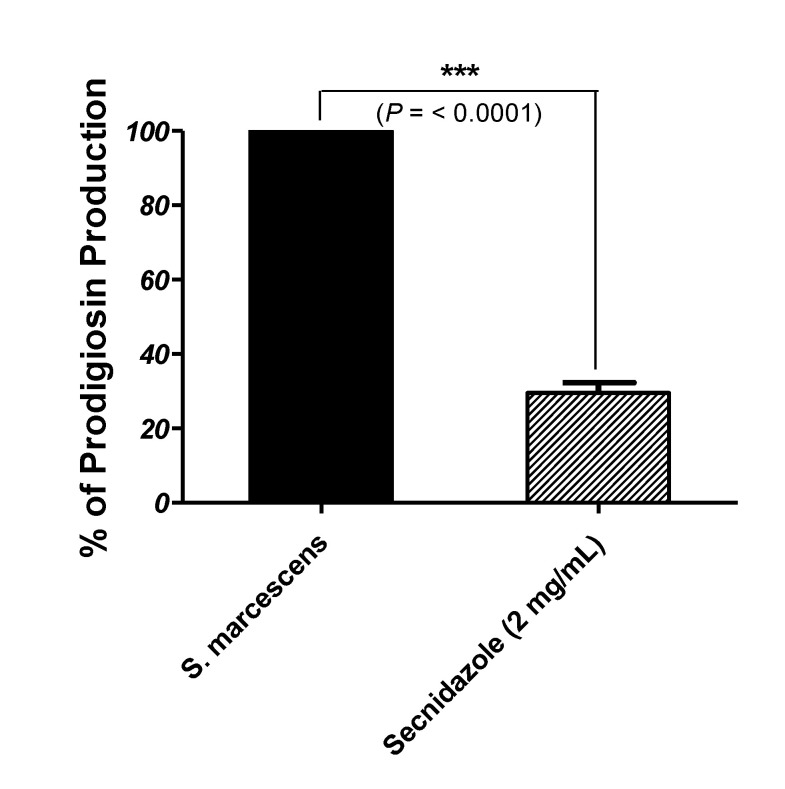
Inhibition of prodigiosin pigment of *S. marcescens* by secnidazole. The significance of the mean difference between secnidazole untreated and treated *S. marcescens* was tested using Student’s *t*-test on absolute values of optical densities, and the result was considered statistically significant when *p* < 0.05. Data are shown as means ± standard error of the percent change of biofilm production by secnidazole at sub-MIC-treated *S. marcescens* from the untreated control. Secnidazole at sub-MIC significantly reduced the prodigiosin pigment production (*p* < 0.0001), *** = *p* < 0.001.

**Figure 5 microorganisms-09-02333-f005:**
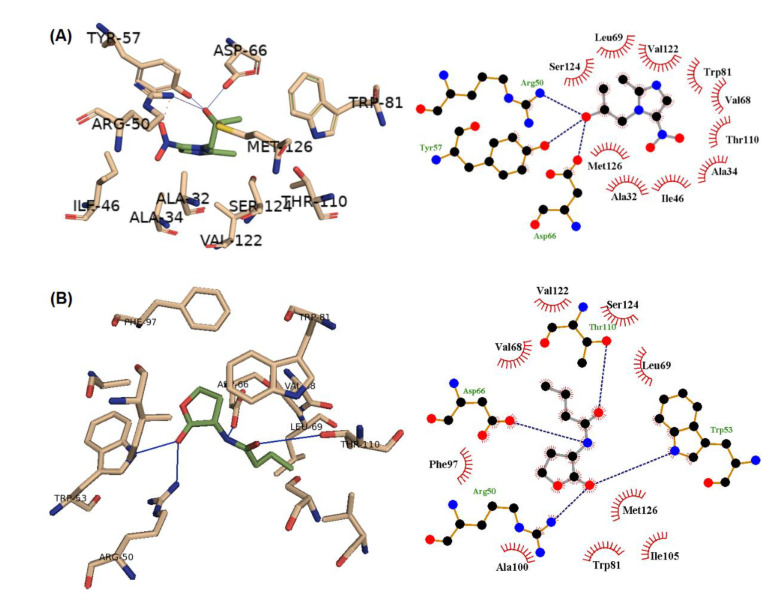
The molecular docking of (**A**) secnidazole and (**B**) C4-HSL into the active site of the SmaR protein: 3D representation (left) and 2D schematic interaction (right); hydrogen bonds (blue lines) and electrostatic interaction (red dashed).

**Figure 6 microorganisms-09-02333-f006:**
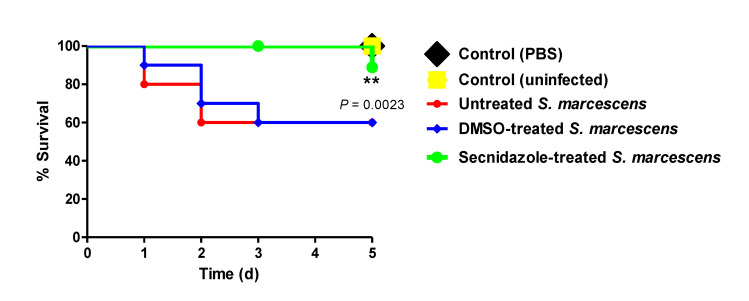
Protection of mice from *S. marcescens* by secnidazole. Five mouse groups comprising 10 mice each were used. The survival of mice in each group was reported every day for 5 days and plotted using the Kaplan–Meier method. The significance (*p* < 0.05) was tested using the log-rank test. All mice in the negative control groups survived, while only 60% of mice survived in the positive control groups. Meanwhile, 90% of mice injected with secnidazole-treated *S. marcescens* survived, showing that secnidazole conferred 30% protection in comparison with mice injected with untreated *S. marcescens*. Log-rank test for trend *p* = 0.0023, ** = *p* < 0.01.

**Table 1 microorganisms-09-02333-t001:** The binding mode of each ligand with the different residues inside the active site of SmaR protein.

Ligand	H-Bonding	Hydrophobic Interaction	AutoDock Score
Secnidazole	Arg 50, Tyr 57, Asp 66 69	Ala32, Ala34, Ile 46, Val 68, Leu 69, Trp 81, Thr110, Val 122, Ser 124, Met 126	−5.3
C4-HSL	Arg 50, Trp 53, Asp 66, Thr 110	Val 68, Leu 69, Trp 81, Phe 97, Ala 100, Ile 105, Val 122, Ser 124, Met 126	−6.4

## Data Availability

Not applicable.

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
