# Peer review of "Secnidazole Is a Promising Imidazole Mitigator of Serratia marcescens Virulence"

_microorganisms, 2021, doi:10.3390/microorganisms9112333_

Round 1
Reviewer 1 Report
The manuscript entitled: “Secnidazole is a promising imidazole mitigator of Serratia marcescens virulence” is an interesting and well conducted work; however presents some limitations.
Major and minor comments:
Line 51. The indicated distribution/prevalence is from which country our countries?
Line 105. Indicate the range tested instead of the concentration values.
Line 130. What was the used cell concentration?
Line 133. What was the used crystal violet concentration?
Merge de sections 2.5.1. and 2.5.2. since the description of the assessment of the biofilm formation for just one strain is unnecessary.
Several controls seem to be missing, namely the control with the compound when the supernatant of the cultures is used (as secnidazole is added indirectly to the assay, for example does the compound presents haemolytic potential?). Further, in section 2.12 the authors refer that used DMSO as solvent, thus this control must be added to all the assays.
The protease inhibition assay is not a really quantitative assessment of protease activity. Adjust the information presented accordingly or introduce a new assay.
Line 166. What was the final cell concentration used in the assay?
Line 174. What is the provenience of the erythrocytes?
Remove figure 1, since it is unnecessary. The authors may add the values in parentheses in the text.
Considering the information provided in the results for each virulence inhibition assay, they may be merged in only one section. Also, the graphics may be organized in one figure.
Furthermore, the figure captions are too descriptive, and possess much information already provided in the materials and methods section.
Line 233. What is measured is not a real optical density but rather an absorbance.
In general, concerning the discussion, the authors provide two large paragraphs that are too descriptive and a little repetitive considering the introduction. This must be shortened.
The discussion must be reformulated to better discuss and integrate the results obtained in this study. Further, as prodigiosin does not have a direct role in virulence or pathogenicity of Serratia marcescens, the relevance of its evaluation should be better discussed. Additionally, the role of secnidazole in QS inhibition could be further explored.
Line 417. Finally…
Author Response
Dear Reviewer 1,
We appreciate the reviewer for your valuable and constructive comments and suggestions, which greatly helped us to improve the manuscript.
Please find the attached word file which includes our replies to you.
Best Regards,
Author

Reviewer 2 Report
The present manuscript describes “Secnidazole is a promising imidazole mitigator of Serratia marcescens virulence”. Secnidazole (nitroimidazole) is an antibiotic that fights bacteria. It has been used in women to treat vaginal bacteria infections. S. marcescens is one of the frequent opportunistic human pathogens that can cause several hospital-acquired infections, for instance pneumonia, intravenous catheter-associated infections, endocarditis, osteomyelitis, and urinary tract infections. S. marcescens developed a considerable to different antibiotics and is equipped with an armory of virulence factors. These virulence factors are regulated in S. marcescens by intercellular communication system Quorum sensing (QS). Targeting bacterial virulence and QS is an interesting approach to mitigate bacterial pathogenesis and overcome the resistance development to antimicrobials. The authors were purchased secindazole from sigma Aldrich. The authors used S.marcescens strain and it was isolated from endotracheal aspiration of an admitted patient in the Intensive Care Unit in Zagazig University Hospital, Zagazig, Egypt. The authours evaluated the anti-virulence activities of secnidazole on a clinical isolate of S. marcescens. The effect of secnidazole at sub-inhibitory concentrations on virulence factors: swarming motility, biofilm formation, protease, hemolysis activity, and prodigiosin production was evaluated in-vitro. The secindidazole protective activity against S. marcescens pathogenesis was in-vivo assessed on mice. The authors found that the secnidazole at sub-MIC significantly reduced the S. marcescens virulence factions production in-vitro and diminished it’s pathogenesis on mice. The authours proved that secnidazole is a promising anti-virulence agent that may be used to control the infections caused S. marcescens. I would recommend to publish this great work in Microorganisms Journal.
Author Response
Dear Reviewer 2,
We are very grateful and highly appreciate the reviewer's nice words, it is our pride. Again, we are very thankful for your deep understanding of our work and its aim.
Best Regards,
Authors
Round 2
Reviewer 1 Report
During the revision some grammar errors were introduced, please revise the document.
Add the origin of the erythrocytes used to the manuscript.
Author Response
Dear Reviewer,
Thank you for your careful revision of our manuscript. We revised the manuscript grammatically and typo errors have been corrected. We added the missed information as indicated. We are appreciating your great efforts to enhance the quality of the manuscript, and for sure we are all ears to make any further modifications.
Best Regards,
Wael